# Genetic Polymorphisms of Nuclear Factor-κB Family Affect the Bone Mineral Density Response to Zoledronic Acid Therapy in Postmenopausal Chinese Women

**DOI:** 10.3390/genes13081343

**Published:** 2022-07-27

**Authors:** Wen-Jie Wang, Jin-Wei He, Wen-Zhen Fu, Chun Wang, Zhen-Lin Zhang

**Affiliations:** Shanghai Clinical Research Center of Bone Disease, Department of Osteoporosis and Bone Disease, Shanghai Jiao Tong University Affiliated Sixth People’s Hospital, 600 Yishan Road, Shanghai 200233, China; wwjiesy@foxmail.com (W.-J.W.); he_jinwei@live.com (J.-W.H.); fu_wenzhen@outlook.com (W.-Z.F.); wangchun66@sjtu.edu.cn (C.W.)

**Keywords:** postmenopausal osteoporosis, single nucleotide polymorphism, NF-κB pathway, bone mineral density

## Abstract

The aim of this study was to explore the allelic association between genetic polymorphisms of the NF-κB pathway and the variance of clinical effects of zoledronic in postmenopausal Chinese women with osteoporosis. In the study, 110 Chinese postmenopausal women with osteoporosis were recruited. Every patient received zoledronic once a year. BMD was measured at baseline and after one year of treatment. The 13 tagger SNPs of five genes in the NF-κB pathway were genotyped. In the study, 101 subjects completed the one-year follow-up. The ITCTG and DTCTG haplotypes, which are constituted of rs28362491, rs3774937, rs230521, rs230510 and rs4648068 of the NF-κB1 gene, were associated with improvement in BMD at L1-4 and femoral neck (*p* < 0.001, *p* = 0.008, respectively). The CGC haplotype, which is constituted of rs7119750, rs2306365 and rs11820062 of the RELA gene, was associated with improvement in BMD at total hip (*p* < 0.001). After Bonferroni correction, haplotypes ITCTG and CGC still showed significant association with the % change of BMD at L1-4 and total hip. Therefore, NF-κB1 and RELA gene were significantly associated with bone response to the treatment of zoledronic in postmenopausal Chinese women with osteoporosis.

## 1. Introduction

Osteoporosis is a common chronic disease characterized by reduced bone strength that predisposes increased risk of fracture. Hip fracture is the most serious complication of osteoporosis, which could seriously threaten physical and psychological health, and diminish quality of life in the elderly population. In Beijing, China, the hip fracture rate in individuals aged 50 years and older increased 2.76-fold in women and 1.61-fold in men from 2002 to 2006, compared with the rates between 1990 and 1992 [1]. The costs of osteoporotic fractures are estimated to double by 2035 and to increase to $25.43 billion by 2050 [2]. Osteoporosis and osteoporotic fractures have become a common problem, whether at individual or socioeconomic levels.

Therapeutic agents for osteoporosis fall into two major categories, antiresorptive drugs and anabolic drugs. Bisphosphonates, such as alendronate, ibandronate, risedronate, and zoledronic acid, are the most widely used antiresorptive drugs, which could decrease bone resorption, increase bone mineral density (BMD) and reduce the risk of fracture. Among them, zoledronic acid, the only once-yearly antiresorptive drug of the bisphosphonate family, has equivalent efficacy to once-weekly oral alendronate and has better compliance and cost-effectiveness compared to alendronate [3,4,5]. Studies have shown that treatment with annual intravenous infusion of zoledronic acid increases BMD and reduces the risk of vertebral fracture by 70%, the risk of hip fracture by 41% and the risk of nonvertebral fracture by 25% during a three-year period [6].

However, therapeutic response to zoledronic acid therapy is variable. Previous studies have shown the correlation between genetic factors and the response to treatment with bisphosphonate. For instance, vitamin D receptor gene (VDR) has been shown to be associated with the response to osteoporosis treatment in postmenopausal women [7,8,9]. In our previous study, we demonstrated the association between the polymorphisms in MVK and FDFT1 and therapeutic response to alendronate therapy in postmenopausal Chinese women [10]. We also explored the relationship between OPG, RANKL, RANK genes and bone response to alendronate, though they did not appear to be linked in the study. Recently, our study has demonstrated the relationship between genetic variants in SOST and therapeutic response to alendronate treatment [11]. Nevertheless, few studies have explored the genetic factors of zoledronic acid therapy.

As we know, bisphosphonate increases BMD and reduces the risk of fracture by inhibiting osteoclast function. So, osteoclast activity is essential for therapeutic effect evaluation. Growing evidence demonstrates that the nuclear factor-κB (NF-κB) pathway plays a crucial role in not only inflammation and tumor promotion [12,13], but also in skeletal homeostasis [14]. The NF-κB family includes NF-κB1 (p50 and its precursor p105), NF-κB2 (p52 and its precursor p100), Rel A (p65), RelB and Rel (c-Rel) [15]. P50/p65 NF-κB heterodimer combines with the inhibitor IκBα in resting cells. When activated by agonists, NF-κB signaling initiates along canonical and non-canonical pathways. In the canonical NF-κB pathway, the IκB kinase complex (IKK), composed of IKKβ and NF-κB essential modulator (NEMO), can phosphorylate and degrade IκBα, leading to the release of p65/p50 dimers and subsequent nuclear translocation [16]. In contrast, in the non-canonical NF-κB pathway, the regulatory key is NF-κB inducing kinase (NIK), which activates IKKα leading to phosphorylation of the inhibitory protein p100 and, then, release of p52 and RelB into the nucleus [17,18].

Growing evidence demonstrates the pivotal role of the NF-κB pathway in osteoclastogenesis and osteoclast differentiation. Mice without both p50 and p52 develop osteopetrosis because of a defect in osteoclast differentiation [19]. Besides, mice without IKKβ have defects in osteoclastogenesis in vitro and vivo [20]. Mice with NIK or RelB knockout possess a hampered ability for bone resorption in models of tumor-induced osteolysis [21]. In addition, Chang et al. found that IKK/NF-κB inhibition reduced bone loss in an animal model of osteoporosis, thus revealing the important significance of NF-κB on osteoblast activity and bone formation [22]. Subsequently, the idea was confirmed in humans. Mutation in the NEMO gene causes osteopetrosis, lymphoedema, hypohidrotic ectodermal dysplasia and immunodeficiency (OL-HED-ID) [23]. In recent research conducted by Frederiksen et al., mutation of the RELA gene was found to be responsible for neonatal osteosclerosis [24].

As we know, alendronate inhibits osteoclast activity by targeting the mevalonate pathway. The NF-κB pathway plays a crucial role in osteoclastogenesis and osteoclast differentiation. Therefore, we assume genetic polymorphisms of nuclear factor-κB family may contribute to variable responses to zoledronic acid treatment. In our study, we chose 13 SNPs in the NF-κB1 gene, NF-κB2 gene, RELA gene, RELB gene and REL gene of the NF-κB pathway to investigate the association between the NF-κB pathway variations and bone response to zoledronic acid therapy in Chinese postmenopausal women with osteoporosis or osteopenia.

## 2. Materials and Methods

### 2.1. Study Subjects

A total of 110 women were recruited from 2014 to 2016 at the department of Osteoporosis and Bone Disease in Shanghai Jiao Tong University Affiliated Sixth People’s Hospital. Inclusion criteria were: (1) One year post-menopause after 40 years of age; (2) body mass index (BMI) between 18 and 30 kg/m^2^; (3) T-score of BMD at the lumbar spine 1–4 (L1–4), the femoral neck or the total hip below −2.5. Exclusion criteria were: (1) chronic hepatic or renal dysfunction or cardio-cerebrovascular disorders; (2) allergy or intolerance to bisphosphonates; (3) presence of other metabolic or inherited bone diseases, such as Paget’s disease of the bone, osteomalacia, hypophosphatasia, hyperparathyroidism etc.; (4) presence of endocrine disease that affects bone metabolism, such as diabetes, thyroid disorder, Cushing syndrome, etc.; (5) treatment history of bisphosphonates, calcitonin, strontium ranelate, denosumab, teriparatide, cathepsin K inhibitors, sclerostin antibodies, a selective estrogen receptor modulatorn or estrogen; (6) treatment history of corticosteroid or anticonvulsants; (7) hypocalcemia or hypophosphatemia. All participants received once-yearly 5 mg intravenous zoledronic acid treatment, 600 mg calcium and 125IU vitamin D3 daily for one year. The study was approved by the Ethics Committee of the Shanghai Jiao Tong University affiliated Sixth People’s Hospital.

### 2.2. Bone Response Assessment

BMD of L1–4, femoral neck or total hip was measured at baseline and after 1 year of treatment by lunar prodigy dual energy X-ray absorptiometry densitometer (GE Healthcare, Madison, WI, USA).

The data were analyzed by the Prodigy enCORE software (ver. 6.70, standard-array mode; GE Healthcare). The coefficient variations (CV) at L1–4, total hip and femoral neck were 1.39, 0.70 and 2.22%, respectively [25]. The least significant change in BMD at L1–4, total hip and femoral neck were 3.85, 1.94 and 6.15%, respectively. Meanwhile, serum calcium (Ca), phosphorus(P), parathyroid hormone (PTH), 25-hydroxyvitamin D [25(OH)D], and β-isomerised C-terminal telopeptide of type 1 collagen (β-CTX), and serum total alkaline phosphatase (ALP) were measured at baseline and after 3 months, 6 months and 12 months of treatment. Serum levels of parathyroid hormone and total serum 25-hydroxyvitamin D were detected by the automatic electrochemiluminescence system (Roche Diagnostic, Mannheim, Germany).

The CV values of serum Ca, P, ALP, PTH, 25(OH)D and β-CTX were 2.4%, 3.3%, 5.1%, 9.5%, 4.0% and 3.4%, respectively [26]. All serum samples were collected between 8 am and 9 am after overnight fasting.

### 2.3. SNP Selection and Genotyping

In the study, we chose 13 SNPs of NF-κB1 gene, NF-κB2 gene, RELA gene, RELB gene, and REL gene of NF-κB pathway from NCBI LocusLink (http://www.Ncbi.nlm.nih.gov/LocusLink/, accessed on 7 March 2017) and HapMap (http://hapmap.ncbi.nlm.nih.gov/, accessed on 7 March 2017) (Table 1). The minor allele frequency (MAF) of SNPs should be higher than 0.05 and the linkage disequilibrium (LD) should be higher than 0.8 (r2 > 0.8). Genomic DNA was isolated from peripheral blood leukocytes via the isopropanol-precipitating method. The quality of the DNA samples was tested by Agarose Gel Electrophoresis. Concentration and purification determinations of the samples were performed by ultraviolet spectrophotometer. The amount of 260 nmOD/280 nmOD of both samples ranged from 1.6–1.8. Genotyping for the polymorphism was performed using the specific primers shown in Table 2. The whole reacting volume of PCR was 20 μL, consisting of 1 μL DNA sample, 1× GC-I buffer (Takara), 3.0 mM Mg^2+^, 0.3 mM dNTP, 1 U HotStarTaq polymerase (Qiagen Inc., Shanghai, China) and 1 µL PCR primer. The reactions were performed on the Mx3000p Real-Time PCR System (Stratagene, La Jolla, CA, USA). Genotyping of genes was performed using the high-throughput SNaPshot technique (Applied Biosystems, Foster City, CA, USA).

Haplotypes were constructed from the population genotypic data by the Stephens algorithm using the Phase program version 2.0.2 (http://www.stat.washington.edu/stephens/phase/download.2.0.2.html, accessed on 10 June 2017). The significance level for LD between tested gene markers was assessed according to the observed haplotype and allelic frequencies using Haploview version 3.2 (http://www.broadinstitute.org/scientific-community/science/programs/medical-and-population-genetics/haploview/haploview, accessed on 10 June 2017).

## 3. Results

### 3.1. Basic Characteristics of Study Subjects

Of the 110 subjects enrolled in our study, 101 subjects completed the study. At baseline, mean age was 66.2 ± 7.82 years, mean height was 152.9 ± 5.89 cm and mean BMI was 22.8 ± 3.28 kg/cm^2^. The mean BMD of L1–4, the femoral neck, total hip was 0.753 ± 0.153, 0.646 ± 0.104, 0.687 ± 0.111 g/cm^2^, respectively (Table 3). No significant differences in age, BMI, and BMD were found at baseline among the women with different genotypes.

### 3.2. Allele Frequencies and Haplotype Structures

From the NF-κB pathway 13 SNPs (NF-κB1, NF-κB2, RELA, RELB and REL) were genotyped. All the SNPs were successfully genotyped and none of the SNPs failed the frequency test (MAF < 0.01). All 13 SNPs were in accordance with HWE. Four haplotype blocks were constructed from the 13 SNPs: one in NF-κB1, one in NF-κB2, one in RELA and one in RELB (Table 4, Figure 1).

### 3.3. Genotype-Treatment Interaction

After 1 year of zoledronic acid treatment, BMD was significantly increased (*p* < 0.001). Change in BMD of L1–4, the femoral neck, the total hip was 5.71 ± 6.21%, 1.39 ± 3.58%, 1.71 ± 4.43%, respectively. No significant associations were found between gene polymorphism in the NF-κB pathway and changes in BMD after 12 months of treatment (*p* > 0.05) (Table 5). Of rs28362491, rs3774937, rs230521, rs230510 and rs4648068 in NF-κB1, the ITCTG haplotype showed a significant association with % change in BMD at L1–4 (*p* < 0.001). The DTCTG haplotype showed a significant association with % change of BMD at femoral neck (*p* = 0.008). The CGC haplotype of rs7119750, rs2306365 and rs11820062 in RELA showed a nominal association with % change in BMD at total hip (*p* < 0.001). Of these, only the ITCTG haplotype in NF-κB1 and the CGC haplotype in RELA remained significantly associated with % change of BMD after Bonferroni correction (*p* = 0.002) (Table 6).

In order to estimate the effectiveness of treatment, individuals were divided into responder(s) groups and non-responder(s) groups according to the least significant change. In subjects, 59.4% (60 cases) in spine BMD, 14.8% (15 cases) in femoral neck, 49.5% (50 cases) in total hip reached an effective therapeutic range. No significant associations were found between gene polymorphism in the NF-κB pathway and BMD response at L1–4, the femoral neck or the total hip after Bonferroni correction (*p* > 0.05) (Table 7). The CGC haplotype in RELA was associated with the BMD response at total hip (*p* = 0.008, OR = 0.397). After Bonferroni correction, however, no significant associations were found between haplotypes and BMD response to zoledronic acid therapy (Table 8).

## 4. Discussion

Bisphosphonate is one of the first line antiresorptive agents commonly used in postmenopausal osteoporosis or osteopenia treatment, which can increase bone mineral density and decrease the risk of fragility fracture [27]. However, as we observed in clinic, the efficacy and safety of bisphosphonate were variable. Previous studies reported that the associations between treatment response to alendronate and genetic variants in the candidate gene of the mevalonate pathway 10, the OPG/RANKL/RANK signaling pathway [28] and the Wnt/β-catenin signaling pathway [29,30,31], indicate that interpatient variability could be due to inherited interindividual differences. However, genetic pharmacological studies about bone response to zoledronic treatment are very limited. Recently, many studies have shown that the nuclear factor-κB (NF-κB) pathway plays a crucial role in the maintenance of osteoclast differentiation and function [14]. In this study, we investigated the relationship between gene polymorphism in the NF-κB pathway and bone response to zoledronic treatment in Chinese osteoporotic women. After one year of treatment with zoledronic, we observed that BMD at L1–4, femoral neck and total hip significantly increased (*p* < 0.001). The ITCTG haplotype in NF-κB1 and the CGC haplotype in RELA were found to correlate with % change of BMD (*p* < 0.001). No correlation was found between gene polymorphism in the NF-κB pathway and changes in BMD after 12 months of treatment (*p* > 0.05). Therefore, we suggest that polymorphisms in NF-κB1 and RELA may be important genetic determinants for the therapeutic response to zoledronic treatment.

NF-κB locates on chromosome 4q23-q24 and plays an important role for multiple diseases associated with inflammation and immunity. The embryo shape of NF-κB1-deficient mice and NF-κB2-deficient mice were normal, with the growth of organs and skeleton in good situation. However, mice without both p50 and p52 developed osteopetrosis because of a defect in osteoclast differentiation [19]. Therefore, we concluded that the NF-κB1 gene and the NF-κB2 gene play an important role in the osteoblast differentiation period, but the mechanisms of action remain mostly unclear, and further studt is necessary [19]. To date, human disease caused by the presence of mutant NF-κB1 gene and NF-κB2 gene have not detected. The NFKB1 rs28362491 polymorphism is an insertion/deletion of four bases in the promoter region of the NFKB1 gene encoding both of the NF-κB1 isoforms, p50 and p105. The allele containing the deletion is less able to bind transcription factors and produces lower transcript levels in luciferase reporter systems. Consequently, carriers of the del-allele have lower levels of NF-κB1 [32,33]. RELA locates on chromosome 11q13 and could encode p65. Animal experiments have shown that high expression of P65 could increase the activity and differentiation of osteoclasts and inhibit the proliferation of osteoblasts [34]. Mutation in the RELA gene has been found responsible for human neonatal osteosclerosis [24]. In this study, we observed that the ITCTG haplotype in NF-κB1 and CGC haplotype in RELA showed a significant association with response to one year of zoledronic acid treatment among patients. However, the roles of NF-κB1 and RELA in bone metabolism are still unclear. Functional study should be performed in the future.

This is the first report on genotype–treatment interaction between NF-κB pathway polymorphisms and response to zoledronic acid therapy, a widely used agent for osteoporosis. We acknowledge that our study has limitations. First, we need a larger sample of postmenopausal women with osteoporosis or osteopenia, which could exclude the impacts of many bias factors. Second, a one-year follow-up might not be enough to detect significant change of BMD. In future, a longer-term follow-up period and more cases are needed.

In conclusion, we observed, for the first time, that polymorphisms in the NF-κB pathway influenced the therapeutic response to zoledronic acid. The ITCTG haplotype in NF-κB1 and the CGC haplotype in RELA were significantly associated with % change in BMD at L1–4 and total hip, respectively.

## Figures and Tables

**Figure 1 genes-13-01343-f001:**
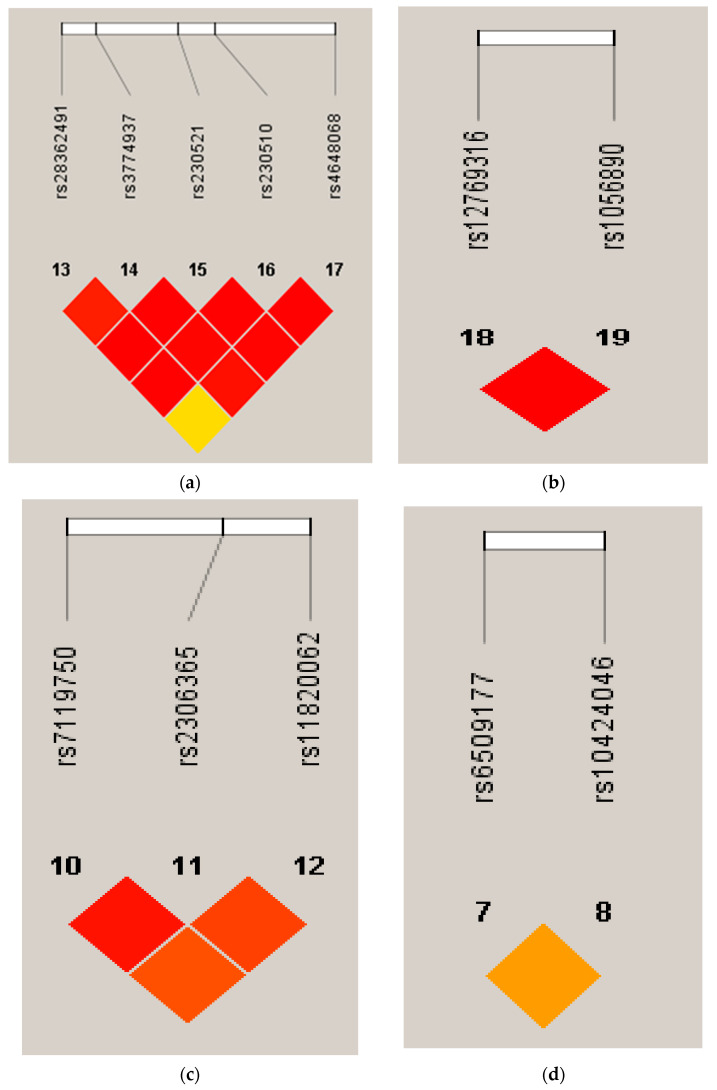
Four haplotype blocks were constructed from 13 SNPs: (**a**) LD plots with R^2^ values of the NF-κB1 gene; (**b**) LD plots with R^2^ values of the NF-κB2 gene; (**c**) LD plots with R^2^ values of the RELA gene; (**d**) LD plots with R^2^ values of the RELB gene.

**Table 1 genes-13-01343-t001:** Information on the 13 SNPs in this study.

SNP	Chr	Position	Gene	SNP Property	Length	Alleles	Major Allele	Minor Allele	MAF	*p*-Value of HWE
rs2306365	11	65427346	RELA	intron5	340	G/A	G	A	0.08	0.47
rs11820062	11	65429936	RELA	intron1	252	C/T	C	T	0.06	0.44
rs7119750	11	65422591	RELA	intron10	150	C/T	C	T	0.07	0.37
rs6509177	19	45530006	RELB	intron6	149	G/A	G	A	0.49	0.43
rs10424046	19	45536036	RELB	intron8	292	G/C	G	C	0.9	0.45
rs12769316	10	104152751	NFκB2	5′FLANKING	215	G/A	G	A	0.18	0.16
rs1056890	10	104162770	NFκB2	3′FLANKING	235	G/A	G	A	0.3	0.2
rs28362491	4	103422155	NFκB1	5′FLANKING	291	C/T	C	T	0.33	0.43
rs230510	4	103476166	NFκB1	intron5	348	A/T	A	T	0.75	0.47
rs230521	4	103463328	NFκB1	intron5	293	G/C	G	C	0.88	0.46
rs4648068	4	103518305	NFκB1	intron14	336	G/A	G	A	0.49	0.39
rs3774937	4	103434253	NFκB1	intron1	218	C/T	C	T	0.96	0.36
rs842647	2	61119471	REL	intron2	179	G/A	G	A	0.59	0.14

SNP single-nucleotide polymorphism, MAF minor allele frequency, HWE Hardy-Weinberg equilibrium.

**Table 2 genes-13-01343-t002:** SNPs studied and their PCR primers.

SNP	Forward	Reverse
rs2306365	ATCAGGAGAGCAGGGGAAGTGG	GCGCATCCAGACCAACAACAAC
rs11820062	AGGGGTGGAGGAAAGGAACCAG	GGGGGAGCAGAGGGAACCTT
rs7119750	GATCCACCCCGTCCTCTGTACC	CCAAGGTGGAGGTGTGGCTAGA
rs6509177	CCCAAGGGAAGAGGCAGCATAA	CCAGACCACCCTGCAACTAGGA
rs10424046	CCGTTTCCTGCCAGAGGACATA	TCTCCCCAAGGCCAAAGACCTA
rs12769316	CAGTGCAGGCCTGTGATTCTGTT	TAGAGGCAGCGAGGTCAGGACA
rs1056890	CATCAAAAGTTCAGGGGCGCTAG	TGAGGTTTAGGGTGGGGAGTGC
rs28362491	GGGCTATGGACCGCATGACTCT	GCACTTCTAGGGGCGCTGTG
rs230510	CAAGCCCCCACACAATAGGAAAA	TTGGCAGGGTTCATCTCCCTAA
rs230521	GCCTGGAGAGGAACCTGGAAAC	AACAGGGTTTTATGCATGGGACA
rs4648068	GCCATTCCCTGACCTTAATGGAAC	AAGCAGAATTGGACACTGGGTCA
rs3774937	CCAAAATGGTACTCAAGGCAGAGG	TGCTGCCTACCATTTCCACATAAA
rs842647	TTCTCTGGGTCATTGACTGATTTGA	TGGGCGACAAGTGTGAAACTCC

**Table 3 genes-13-01343-t003:** Basic characteristics of the 101 postmenopausal women.

Characteristics	Mean ± SD
Age (years)	66.2 ± 7.82
Height (cm)	152.9 ± 5.89
Weight (kg)	53.3 ± 8.14
BMI (kg/cm^2^)	22.8 ± 3.28
Lumbar (L1–4) BMD (g/cm^2^)	0.753 ± 0.153
Femoral neck BMD (g/cm^2^)	0.646 ± 0.104
Total hip BMD (g/cm^2^)	0.687 ± 0.111

Data presented as mean ± SD BMD, Bone mineral density.

**Table 4 genes-13-01343-t004:** Four haplotype blocks and 17 haplotypes of 13 SNPs in NF-kB pathway.

Gene	SNPs	HAP
*RELB*	rs6509177	GC
	rs10424046	AC
		GG
		AG
*RELA*	rs7119750	CGT
	rs2306365	TAC
	rs11820062	CGC
*NFKB1*	rs28362491	DCCTG
	rs230510	ICCTG
	rs230521	DTCTG
	rs3774937	ITCTG
	rs4648068	ITGAA
		DTCTA
		ITGTA
*NFKB2*	rs12769316	GA
	rs1056890	AG
		GG

**Table 5 genes-13-01343-t005:** The associations between 13 SNPs and the % change in BMD.

Gene	SNP	Genetype	Counts	L1–4 (%)			Femoral Neck (%)			Total Hip (%)		
				Mean	SD	*p*-Value	Mean	SD	*p*-Value	Mean	SD	*p*-Value
*REL*	rs842647	G/G	76	6.63	6.33	0.64	1.33	3.5	0.61	1.61	4.34	0.69
		G/A	23	3.3	5.83		1.11	3.79		1.73	5.25	
		A/A	2	5.65	3.07		2.06	2.64		3.68	0.16	
*NFκB1*	rs28362491	ATTG /ATTG	30	5.48	4.59	0.13	2.53	3.77	0.06	1.85	3.06	0.9
		-/-	19	6.93	7.89		0.64	2.28		1.87	5.1	
		-/ATTG	52	5.6	6.35		1.01	3.76		1.56	4.88	
	rs3774937	T/T	37	5.11	4.86	0.13	1.72	4.13	0.24	1.37	4.37	0.87
		C/T	49	5.98	6.16		1.48	3.5		2.07	4.14	
		C/C	15	6.94	8.94		0.3	1.99		1.33	5.57	
	rs230521	G/G	26	5.49	4.8	0.73	2.13	3.85	0.28	1.7	3.26	0.93
		C/G	53	5.55	6.31		1.18	3.85		1.61	4.83	
		C/C	22	6.82	7.45		1.04	2.41		1.94	4.77	
	rs230510	T/T	28	6.16	6.73	0.77	1.17	2.2	0.85	1.86	4.66	0.81
		A/T	54	5.65	6.34		1.53	4.16		1.78	4.65	
		A/A	19	5.8	5.27		1.33	3.6		1.26	3.5	
	rs4648068	A/A	31	5.01	5.16	0.7	1.93	3.98	0.27	1.52	4.08	0.95
		G/A	52	5.94	6.02		1.26	3.72		1.89	4.44	
		G/G	18	6.8	8.26		0.85	2.28		1.5	5.13	
*NFκB2*	rs12769316	G/G	74	5.04	5.46	0.15	1.29	3.47	0.48	1.39	4.29	0.22
		G/A	26	8.06	7.71		1.6	4		2.5	4.83	
		A/A	1	3.58	\		3.61	\		4.19	\	
	rs1056890	G/G	60	6.28	5.87	0.81	1.34	3.61	0.81	1.89	3.8	0.45
		G/A	35	5.02	7.05		1.61	3.46		1.71	5.55	
		A/A	6	5.85	4.59		0.69	4.5		-0.14	2.83	
*RELA*	rs7119750	C/C	41	5.67	6.09	0.66	1.17	3.52	0.53	1.09	3.92	0.22
		C/T	43	6.34	6.76		1.61	3.54		2.09	4.87	
		T/T	17	5	5.4		1.37	4.02		2.2	4.49	
	rs2306365	G/G	39	7.39	5.27	0.65	1.21	4.13	0.45	1.64	4	0.16
		G/A	45	5.59	7.19		1.68	3.09		2.08	4.93	
		A/A	17	2.92	4.47		1.06	3.62		0.87	4.07	
	rs11820062	C/C	40	4.65	5.2	0.61	0.71	3.73	0.31	0.3	4.54	0.03
		C/T	42	6.52	7.11		1.93	3.72		2.6	4.71	
		T/T	19	6.77	6.07		1.65	2.82		2.69	2.52	
*RELB*	rs6509177	A/A	29	5.83	6.46	0.9	1.42	3.42	0.27	1.2	4.32	0.84
		G/A	54	5.39	4.59		1.79	3.43		2.13	3.92	
		G/G	18	7.06	9.44		0.17	4.19		1.24	5.95	
	rs10424046	G/G	30	5.53	6.33	0.79	1.44	3.47	0.28	1.28	4.6	0.91
		G/C	49	5.45	4.67		1.85	3.26		2.18	3.83	
		C/C	22	7	8.69		0.3	4.3		1.23	5.44	

BMD Bone mineral density, SNP single-nucleotide polymorphism.

**Table 6 genes-13-01343-t006:** The associations between 17 haplotypes and the % change in BMD.

Gene	SNPs	HAP	L1–4		Femoral Neck		Total Hip	
			β	*p*-Value	β	*p*-Value	β	*p*-Value
*RELB*	rs6509177	GC	0.17	0.925	−0.51	0.33	0.07	0.91
	rs10424046	AC	2.15	0.643	−0.41	0.75	0.03	0.98
		GG	0.46	0.937	−0.53	0.75	0.56	0.79
		AG	−0.54	0.765	0.61	0.23	−0.13	0.83
*RELA*	rs7119750	CGT	0.94	0.594	0.46	0.37	1.29	0.04
	rs2306365	TAC	−0.77	0.662	0.31	0.54	0.76	0.22
	rs11820062	CGC	−0.2	0.919	−1.02	0.07	−2.64	**0.00008**
*NFκB1*	rs28362491	DCCTG	2.64	0.146	−0.75	0.15	0.02	0.97
	rs3774937	ICCTG	1.29	0.887	3.89	0.13	1.93	0.55
	rs230521	DTCTG	3.58	0.689	−6.68	0.008	−7.43	0.02
	rs230510	ITCTG	−21.6	0.0001	2.54	0.12	0.59	0.77
	rs4648068	ITGAA	−0.55	0.768	0.09	0.75	−0.15	0.82
		DTCTA	−0.35	0.931	0.35	0.76	1.02	0.48
		ITGTA	−0.48	0.909	1.96	0.08	−0.13	0.93
*NFκB2*	rs12769316	GA	0.52	0.81	0.14	0.81	−0.55	0.46
	rs1056890	AG	3.77	0.156	0.53	0.49	1.15	0.22
		GG	−2.29	0.224	−0.37	0.49	0.12	0.85

BMD Bone mineral density, β-CTX β-isomerised C-terminal telopeptide of type 1 collagen, SNPsingle-nucleotide polymorphism, β regression coefficient. Significant association after Bonferroni correction values are shown in bold.

**Table 7 genes-13-01343-t007:** The associations between 13 SNPs and BMD response.

Genes	SNP	L1–4			Femoral Neck			Total Hip		
		OR	95% CI	*p*-Value	OR	95% CI	*p*-Value	OR	95% CI	*p*-Value
*REL*	rs842647	0.846	0.373–1.919	0.688	0.695	0.080–6.033	0.741	1.536	0.669–3.527	0.312
*NFκB1*	rs28362491	1.207	0.673–2.165	0.527	0.096	0.011–0.852	0.035	0.818	0.461–1.452	0.493
*NFκB1*	rs3774937	1.234	0.686–2.22	0.482	0.248	0.045–1.368	0.11	0.926	0.522–1.642	0.792
*NFκB1*	rs230521	1.129	0.632–2.018	0.681	0.18	0.033–0.955	0.044	1.002	0.567–1.771	0.993
*NFκB1*	rs230510	0.793	0.434–1.445	0.449	2.711	0.667–11.01	0.163	0.802	0.445–1.447	0.464
*NFκB1*	rs4648068	1.372	0.759–2.482	0.295	0.213	0.039–1.174	0.076	1.048	0.590–1.86	0.873
*NFκB2*	rs12769316	2.378	0.945–5.982	0.066	0.425	0.044–4.106	0.46	2.083	0.868–4.994	0.1
*NFκB2*	rs1056890	0.74	0.381–1.439	0.375	4.046	0.944–17.35	0.06	0.756	0.391–1.46	0.404
*RELA*	rs7119750	1.034	0.59–1.812	0.907	1.822	0.521–6.371	0.348	1.209	0.694–2.107	0.503
*RELA*	rs2306365	1.118	0.634–1.972	0.701	1.813	0.516–6.366	0.353	1.321	0.751–2.323	0.334
*RELA*	rs11820062	1.551	0.881–2.732	0.128	1.086	0.331–3.561	0.891	1.635	0.937–2.853	0.083
*RELB*	rs6509177	1.239	0.683–2.247	0.481	1.509	0.393–5.787	0.549	1.134	0.633–2.03	0.673
*RELB*	rs10424046	1.316	0.749–2.311	0.34	1.945	0.575–6.575	0.284	0.999	0.577–1.73	0.998

BMD Bone mineral density, SNP single-nucleotide polymorphism, OR odds ratio, CI confidence interval.

**Table 8 genes-13-01343-t008:** The associations between 17 blocks and the % change in BMD response.

Gene	SNPs	HAP	L1–4		Femoral Neck		Total Hip	
			OR	P	OR	P	OR	P
*RELB*	rs6509177	GC	1.18	0.571	1.77	0.392	1.09	0.769
	rs10424046	AC	2.48	0.287	2.55	0.448	0.566	0.456
		GG	1.46	0.693	0.59 × 10^9^	0.999	1.43	0.704
		AG	0.722	0.271	0.577	0.392	0.967	0.908
*RELA*	rs7119750	CGT	1.47	0.192	1.09	0.883	1.54	0.134
	rs2306365	TAC	1.03	0.907	1.82	0.348	1.21	0.503
	rs11820062	CGC	0.547	0.066	0.329	0.283	0.397	**0.00859**
*NFKB1*	rs28362491	DCCTG	1.23	0.477	0.12	0.0612	0.857	0.594
	rs3774937	ICCTG	0.892	0.937	8.27	0.172	3.75 × 10³	0.843
	rs230521	DTCTG	5.22 × 10^3^	0.862	0.01	0.989	0.22 × 10^5^	0.914
	rs230510	ITCTG	0.549	0.525	0.12 × 10^9^	0.999	4.15	0.214
	rs4648068	ITGAA	0.793	0.449	2.71	0.163	0.802	0.464
		DTCTA	0.594	0.426	0.94 × 10^9^	0.998	1.23	0.744
		ITGTA	1.08	0.905	5.19	0.107	1.82	0.37
*NFKB2*	rs12769316	GA	0.74	0.375	4.05	0.0599	0.756	0.405
	rs1056890	AG	2.38	0.066	0.425	0.46	2.08	0.1
		GG	0.855	0.604	0.447	0.236	0.884	0.676

BMD Bone mineral density, SNP single-nucleotide polymorphism, OR odds ratio. Significant association after Bonferroni correction values are shown in bold.

## Data Availability

The data presented in this study are available on request from the corresponding author. The data are not publicly available due to privacy.

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
