# Peer review of "Genetic Polymorphisms of Nuclear Factor-κB Family Affect the Bone Mineral Density Response to Zoledronic Acid Therapy in Postmenopausal Chinese Women"

_genes, 2022, doi:10.3390/genes13081343_

Round 1

Reviewer 1 Report

This polymorphism association study of NF-KB and anti-resorptive drug Zoledronic acid is a typical gene and phenotype association study. Overall, the study is well designed with new finding of association between NF-κB1 haplotype ITCTG and DTCTG and improvement of BMD of spine L1-4 and femur neck after this drug treatment. They also found association between RELA CGC haplotype and total hip BMD improvement. The manuscript is generally well written.  Following are the minor comments need to be addressed.

  1. Did the authors analyze the SNP polymorphism run a agarose gel? Or QPCR? If so, showing a data image will be more convincing.
  2. Why the authors provided the CV values of serum bone metabolism markers and not present the mean values and did not provide any changes before and after treatment?
  3. Line 34: Non-trivial is not a common word.
  4. All Reference has problem the entire manuscript. Make it confusion. Please format according to the Journal.
  5. Line 109 : typos “Meatured “should be measured
  6. Lines 114-115 should spell out every percentage, not like only for last one.
  7. Line 177: “Arrived an effective therapeutic range” what does this sentence mean? I believe arrive should be replaced with reach. This is not a standard English usage here.  
  8. Line 199: Variability should be variable.
  9. Line 105, Manifested is weird English use here. Maybe “demonstrated” or “showed”.
  10. Line 209: “Mice” should be “mice”
  11. Lines 171 and 233 “ What does the author mean saying” Normally association with % change” since P <0.001"
  12. Line 237-238 the meaning of the sentence is not clear. The “one” should be deleted.

Author Response

  1. Did the authors analyze the SNP polymorphism run a agarose gel? Or QPCR? If so, showing a data image will be more convincing.

Thank you for your question. Genotyping for the polymorphism was performed using the specific primers shown in Table 1. The whole reacting volume of PCR was 20μl, consisting of 1 μl DNA sample, 1x GC-I buffer(Takara), 3.0 mM Mg2+, 0.3 mM dNTP, 1 U HotStarTaq polymerase (Qiagen Inc) and 1µl PCR primer. The reactions were performed on the Mx3000p Real-Time PCR System (Stratagene, La Jolla, CA, USA). Genotyping of genes was performed using the high-throughtput SNaPshot technique (Applied Biosystems, Foster City, CA, USA).

Table 1  SNPs studied and their PCR primers

SNP

Forward

Reverse

rs2306365

ATCAGGAGAGCAGGGGAAGTGG

GCGCATCCAGACCAACAACAAC

rs11820062

AGGGGTGGAGGAAAGGAACCAG

GGGGGAGCAGAGGGAACCTT

rs7119750

GATCCACCCCGTCCTCTGTACC

CCAAGGTGGAGGTGTGGCTAGA

rs6509177

CCCAAGGGAAGAGGCAGCATAA

CCAGACCACCCTGCAACTAGGA

rs10424046

CCGTTTCCTGCCAGAGGACATA

TCTCCCCAAGGCCAAAGACCTA

rs12769316

CAGTGCAGGCCTGTGATTCTGTT

TAGAGGCAGCGAGGTCAGGACA

rs1056890

CATCAAAAGTTCAGGGGCGCTAG

TGAGGTTTAGGGTGGGGAGTGC

rs28362491

GGGCTATGGACCGCATGACTCT

GCACTTCTAGGGGCGCTGTG

rs230510

CAAGCCCCCACACAATAGGAAAA

TTGGCAGGGTTCATCTCCCTAA

rs230521

GCCTGGAGAGGAACCTGGAAAC

AACAGGGTTTTATGCATGGGACA

rs4648068

GCCATTCCCTGACCTTAATGGAAC

AAGCAGAATTGGACACTGGGTCA

rs3774937

CCAAAATGGTACTCAAGGCAGAGG

TGCTGCCTACCATTTCCACATAAA

rs842647

TTCTCTGGGTCATTGACTGATTTGA

TGGGCGACAAGTGTGAAACTCC

We added a relevant figure in revised version.

  1. Why the authors provided the CV values of serum bone metabolism markers and not present the mean values and did not provide any changes before and after treatment?

Thank you for your suggestions. It’s a pity that the in-depth analysis of serum bone metabolism markers changes before and after treatment is very difficult because of a lack of available data. The CV values originated from our previous studies.

  1. Line 34: Non-trivial is not a common word.
  2. All Reference has problem the entire manuscript. Make it confusion. Please format according to the Journal.
  3. Line 109 : typos “Meatured “should be measured
  4. Lines 114-115 should spell out every percentage, not like only for last one.
  5. Line 177: “Arrived an effective therapeutic range” what does this sentence mean? I believe arrive should be replaced with reach. This is not a standard English usage here.
  6. Line 199: Variability should be variable.
  7. Line 105, Manifested is weird English use here. Maybe “demonstrated” or “showed”.
  8. Line 209: “Mice” should be “mice”
  9. Lines 171 and 233 “ What does the author mean saying” Normally association with % change” since P <0.001"
  10. Line 237-238 the meaning of the sentence is not clear. The “one” should be deleted.

We are very grateful for your serious and carefulness to our study. We have corrected in revised version.

Reviewer 2 Report

Genetic polymorphisms of nuclear factor-κB family affect the bone mineral density response to zoledronic acid therapy in postmenopausal Chinese women tried to genotype the polymorphism of NFkB family and figure out the relationship between these SNPs and BMD after 1 year treatment. The authors didn't detect any correlation between them, but detected some correlation between haplotype and the treatment. 

The manuscript needs to be carefully revised. The serial number of the citations is totally mixed with the main text, especially the numbers and years in the main text. Please switch the serial No. to superscript.

Genotyping is mentioned multiple times in the text. The authors need to describe the genotyping in the Method. It is important for the understanding of this research.

The authors discussed the detail of rs28362891 on NFkB, however the paper didn’t explain the rational to choose the other SNPs in NF-kB pathways and what the SNPs represent in pathology of osteoporosis, especially considering that many of them locates in intron not exon. Please discuss briefly how SNPs in intron affect NF-kB pathway.

Figure 1 is not properly described in the paper, so it is hard to understand the meaning of the content in figure 1. Why is the haplotype in this research important? How did the authors recognize the haplotype of SNPs? It should be discussed in Method. How does it affect the correlation between SNPs and the change of BMD? 

Author Response

  1. The manuscript needs to be carefully revised. The serial number of the citations is totally mixed with the main text, especially the numbers and years in the main text. Please switch the serial No. to superscript.

We are very grateful for your serious and carefulness to our study. We have corrected in revised version.

  1. Genotyping is mentioned multiple times in the text. The authors need to describe the genotyping in the Method. It is important for the understanding of this research.

Thank you for your question. Genotyping for the polymorphism was performed using the specific primers shown in Table 1. The whole reacting volume of PCR was 20μl, consisting of 1 μl DNA sample, 1x GC-I buffer(Takara), 3.0 mM Mg2+, 0.3 mM dNTP, 1 U HotStarTaq polymerase (Qiagen Inc) and 1µl PCR primer. The reactions were performed on the Mx3000p Real-Time PCR System (Stratagene, La Jolla, CA, USA). Genotyping of genes was performed using the high-throughtput SNaPshot technique (Applied Biosystems, Foster City, CA, USA).

Table 1  SNPs studied and their PCR primers

SNP

Forward

Reverse

rs2306365

ATCAGGAGAGCAGGGGAAGTGG

GCGCATCCAGACCAACAACAAC

rs11820062

AGGGGTGGAGGAAAGGAACCAG

GGGGGAGCAGAGGGAACCTT

rs7119750

GATCCACCCCGTCCTCTGTACC

CCAAGGTGGAGGTGTGGCTAGA

rs6509177

CCCAAGGGAAGAGGCAGCATAA

CCAGACCACCCTGCAACTAGGA

rs10424046

CCGTTTCCTGCCAGAGGACATA

TCTCCCCAAGGCCAAAGACCTA

rs12769316

CAGTGCAGGCCTGTGATTCTGTT

TAGAGGCAGCGAGGTCAGGACA

rs1056890

CATCAAAAGTTCAGGGGCGCTAG

TGAGGTTTAGGGTGGGGAGTGC

rs28362491

GGGCTATGGACCGCATGACTCT

GCACTTCTAGGGGCGCTGTG

rs230510

CAAGCCCCCACACAATAGGAAAA

TTGGCAGGGTTCATCTCCCTAA

rs230521

GCCTGGAGAGGAACCTGGAAAC

AACAGGGTTTTATGCATGGGACA

rs4648068

GCCATTCCCTGACCTTAATGGAAC

AAGCAGAATTGGACACTGGGTCA

rs3774937

CCAAAATGGTACTCAAGGCAGAGG

TGCTGCCTACCATTTCCACATAAA

rs842647

TTCTCTGGGTCATTGACTGATTTGA

TGGGCGACAAGTGTGAAACTCC

We added a relevant figure in revised version.

  1. The authors discussed the detail of rs28362891 on NFkB, however the paper didn’t explain the rational to choose the other SNPs in NF-kB pathways and what the SNPs represent in pathology of osteoporosis, especially considering that many of them locates in intron not exon. Please discuss briefly how SNPs in intron affect NF-kB pathway.

Thank you for your question. Although there are quite many correlation researches regarding to 28362891 on NFkB, the researches of the other SNPs in NF-kB pathway are limited. The functional study should be performed in the future.

  1. Figure 1 is not properly described in the paper, so it is hard to understand the meaning of the content in figure 1. Why is the haplotype in this research important? How did the authors recognize the haplotype of SNPs? It should be discussed in Method. How does it affect the correlation between SNPs and the change of BMD?

Thank you for your question. Haplotypes were constructed from the population genotypic data by the Stephens algorithm using the Phase program version 2.0.2 (http://www.stat.washington.edu/stephens/phase/download.2.0.2.html). The significance level for LD between tested gene markers was assessed according to the observed haplotype and allelic frequencies using Haploview version 3.2 (http://www.broadinstitute.org/scientific-community/science/programs/medical-and-population-genetics/haploview/haploview). Four haplotype blocks have been constructed from 13 SNPs: (a) LD plots with R2 values of the NF-κB1 gene; (b) LD plots with R2 values of the NF-κB2 gene; (c) LD plots with R2 values of the RELA gene; (d) LD plots with R2 values of the RELB gene, as shown in Table 2.

Table 2 Four haplotype blocks and 17 haplotypes of 13 SNPs in NF-kB pathway

Gene

SNPs

HAP

RELB

rs6509177

GC

rs10424046

AC

GG

AG

RELA

rs7119750

CGT

rs2306365

TAC

rs11820062

CGC

NFKB1

rs28362491

DCCTG

rs230510

ICCTG

rs230521

DTCTG

rs3774937

ITCTG

rs4648068

ITGAA

DTCTA

ITGTA

NFKB2

 rs12769316

GA

rs1056890

AG

GG

We added a relevant figure in revised version.

Reviewer 3 Report

None.

Author Response

Thank you for your time and consideration. 

Round 2

Reviewer 2 Report

The revised version addresses all my concerns.